# Alagille Syndrome: Diagnostic Challenges and Advances in Management

**DOI:** 10.3390/diagnostics10110907

**Published:** 2020-11-06

**Authors:** Mohammed D. Ayoub, Binita M. Kamath

**Affiliations:** 1Division of Gastroenterology, Hepatology, and Nutrition, The Hospital for Sick Children, University of Toronto, 555 University Avenue, Toronto, ON M5G 1X8, Canada; mohammed.ayoub@sickkids.ca; 2Department of Pediatrics, Faculty of Medicine, Rabigh Branch, King Abdulaziz University, P.O. Box 80205, Jeddah 21589, Saudi Arabia

**Keywords:** Alagille syndrome, bile duct paucity, *JAG1*, *NOTCH2*, intestinal bile acid transporters

## Abstract

Alagille syndrome (ALGS) is a multisystem disease characterized by cholestasis and bile duct paucity on liver biopsy in addition to variable involvement of the heart, eyes, skeleton, face, kidneys, and vasculature. The identification of *JAG1* and *NOTCH2* as disease-causing genes has deepened our understanding of the molecular mechanisms underlying ALGS. However, the variable expressivity of the clinical phenotype and the lack of genotype-phenotype relationships creates significant diagnostic and therapeutic challenges. In this review, we provide a comprehensive overview of the clinical characteristics and management of ALGS, and the molecular basis of ALGS pathobiology. We further describe unique diagnostic considerations that pose challenges to clinicians and outline therapeutic concepts and treatment targets that may be available in the near future.

## 1. Introduction

Alagille syndrome (ALGS) is an inherited multi-organ disease of variable severity. The first clinical description of ALGS was made by the French hepatologist Daniel Alagille in 1969 who reported on 30 patients with hypoplastic intra-hepatic bile ducts, of which 50% appeared to have readily recognizable extrahepatic clinical features [1]. It was not until decades later that a deeper understanding of the variability and severity of the clinical phenotype and mode of inheritance was appreciated. In 1987, Alagille reported on a larger series of 80 children with paucity of intra-hepatic bile ducts associated with variable degrees of chronic cholestasis, characteristic facial features, structural heart disease, posterior embryotoxon, and vertebral arch defects [2,3]. Due to the absence of advanced molecular diagnostics, the diagnosis of ALGS was established with presence of three out of the five features above in addition to bile duct paucity on liver biopsy. The mode of inheritance was deemed to be autosomal dominant with variable penetrance due to the identification family members with isolated anomalies [2].

The incidence of clinically apparent ALGS is approximately 1 in 70,000 live births, but this was estimated based on the presence of neonatal cholestasis in the pre-molecular diagnostics era [4]. However, following the discovery that mutations in *JAGGED1 (JAG1)* are responsible for ALGS, through screening of relatives of ALGS mutation positive probands (of which 47% did not meet clinical criteria), the true incidence is likely 1 in 30,000 live births [5,6,7].

The purpose of this article is to provide a broad clinical overview of ALGS, with a specific focus on diagnostic challenges to gastroenterologists and pathologists, as well as current and future approaches to the management of patients with ALGS.

## 2. Clinical Overview

The classic descriptions of ALGS report potential involvement of five organ systems (liver, face, eye, heart and skeleton). It is important to note that the pattern and degree of organ involvement may be different among patients, even those in the same family sharing the same mutation [2,7,8,9,10,11,12]. Following these initial reports [2,8], several larger descriptive studies consistently showed a significant degree of renal and vascular involvement [9,10,11,13]. Therefore, ALGS clinical criteria have been expanded to include seven instead of five main organ systems, which, in the absence of a molecular diagnosis or family history, requires involvement of at least three organs for diagnosis [7,13,14].

### 2.1. Hepatic Features

The liver is the classically involved organ in ALGS with a frequency of 89–100% in cohorts ascertained from gastroenterologists [2,8,9,10,11,12]. Cholestasis is usually evident in the first year of life, and many infants are evaluated in the first few weeks after birth for conjugated hyperbilirubinemia and scleral icterus. In a review by Subramaniam et al. on 117 ALGS patients, where the majority were diagnosed prior to 1 year of age, jaundice was found in 89% [12]. Hepatomegaly is evident in 70–100% of ALGS patients [2,12].

Synthetic liver function is typically preserved especially in the first year of life. Coagulopathy in this period is likely due to fat-soluble vitamin deficiency (FSVD) from severe cholestasis leading to vitamin K deficiency, rather than liver dysfunction, and is easily corrected with supplementation. Splenomegaly is quite uncommon early in the disease course, but is present in 70% of patients with advancing age and represents fibrosis and evolving portal hypertension [10]. 

Although FSVD can result in numerous complications, such as bleeding, increased risk of fractures, and growth failure, the most debilitating symptom of cholestasis in ALGS is intense pruritus, which is amongst the worst of any cholestatic liver disease. It is associated with elevated serum bile salt levels and may or may not associated with jaundice (anicteric pruritus). Significant pruritus becomes apparent at around 6 months of age, causing skin disfigurement from excoriations and sleep disruption [12,14,15]. Scratch marks are usually visible on the ears, trunk, and feet.

Additional consequences of cholestasis include the development of cutaneous xanthomas as a result of hypercholesteremia (Figure 1B,C). These typically painless lesions appear on extensor surfaces of the hands, knees, and inguinal creases, and correspond to a serum cholesterol > 500 mg/dL [14,16]. They improve with resolution of cholestasis during childhood and also invariably disappear after liver transplantation (LT) [14,15,16].

The hepatic prognosis in ALGS was previously regarded as favorable, with reportedly only 20–30% requiring LT [10,17]. However, these data represent mixed cohorts of children with ALGS, those with and without significant liver disease. Kamath et al. recently reported on the outcomes of 293 ALGS patients with cholestasis in a prospective observational multicenter North American study [18]. Native liver survival in ALGS probands with cholestasis was only 24% at 18.5 years. In early childhood, LT in ALGS typically occurs due to complications of cholestasis and this study revealed an additional burden of liver disease in later childhood due to fibrosis and portal hypertension [10,11,17].

### 2.2. Cardiac Features

Structural cardiac disease is a cause of great morbidity and mortality in patients with ALGS [10]. In the largest cohort of ALGS to date evaluating cardiac phenotype in 200 patients, cardiac involvement was present in 94%, with predominance of right sided anomalies [19]. The most commonly reported lesions confirmed on imaging or detectable with a murmur include branch pulmonary artery stenosis or hypoplasia (76%), Tetralogy of Fallot (TOF) (12%), and left-sided lesions, such as valvular and supravalvular aortic stenosis (7%). TOF in ALGS tends to be more severe and is more likely associated with pulmonary atresia in comparison with the general population (35% vs. 20%) [20].

Complex cardiac disease is responsible for early death in 15% of patients with ALGS and is associated with a predicted 6-year-survival of only 40% [10]. The highest mortality rate was reported at 75% in patients with TOF and pulmonary atresia, and 34% in TOF alone [10].

### 2.3. Facial Features

Distinctive facies in patients with ALGS is a highly penetrant feature, though can be difficult to appreciate in infants [2]. It is present in 70–96% of ALGS patients [2,8,10]. Features include an inverted triangular face formed by a high prominent forehead and a pointed chin, deep set eyes and hypertelorism, and a straight nose with a bulbous tip (Figure 1A) [7,14,21].

### 2.4. Ocular Features

Numerous ocular abnormalities have been reported in ALGS, of which posterior embryotoxon is the most common feature reported in 56–95% of patients [8,22]. It is prominence of the lines of Schwalbe and is detected by slit lamp examination in the anterior chamber of the eye. It is not pathognomonic of ALGS as it is seen in 22% of the general population and up to 70% of patients with 22q11 syndrome [23,24]. Since posterior embryotoxon is of no visual consequence, the utility of its presence is mainly to aid the clinical diagnosis of ALGS. 

Another ophthalmologic feature of ALGS that has been described is optic disk drusen. This is identified with ocular ultrasound and in one study was detected in one eye in 95% and both eyes in 80% of 20 ALGS patients, compared to none of 8 non-ALGS cholestatic patients [25]. As the prevalence in the general population is approximately 0.4-2.4%, and is much lower than that of posterior embryotoxon [26], it may prove to be specific and valuable finding to aid in the clinical diagnosis of ALGS.

### 2.5. Skeletal Features

Skeletal involvement in ALGS is highly variable, ranging from inconsequential vertebral anomalies to osteopenia with pathological fractures [2,27,28]. The most commonly reported anomaly is butterfly vertebrae seen in 33–87%, formed by incomplete fusion of the anterior arch [2,12]. They may be present in other genetic syndromes, such as Jarcho–Levin syndrome, Kabuki syndrome, the Vertebral defects, Anal atresia, Cardiac defects, Tracheo-esophageal fistua, Renal anomalies, and Limb abnormalities (VACTERL) association, and other causes of cholestasis [29,30]. Sanderson et al. found a significantly higher frequency of vertebral anomalies in patients with ALGS, compared to patients with other causes of cholestasis (66% vs. 10%) [31]. In addition, extremity abnormalities are also common in ALGS. Kamath et al. reported increased presence of supernumerary digital flexion creases in ALGS compared to general population (35% vs. <1%), which may aid in clinical diagnosis [32]. Other reported abnormalities include shortened distal phalanges and fifth finger clinodactyly [10], and bilateral radio-ulnar synostosis [33].

Pathological fractures are common in patients with ALGS. A survey study indicated that 28% of ALGS patients have experienced a fracture, with more than two-thirds of fractures involving the lower extremities, and associated with an unimpressive mechanism of injury, if any [28]. A more recent study evaluated bone mineral density and content (BMD and BMC) of children > 5 years with ALGS compared to other causes of cholestasis using Dual-energy X-ray absorptiometry (DEXA) scans [27]. This study found significantly lower Z-scores in ALGS, but these normalized after adjusting for anthropometrics. However, Z-scores correlated negatively with serum bile acids and total bilirubin in patients with previous fractures despite adjusting for weight and height. To adequately assess cortical and trabecular bone separately, Kindler et al. utilized peripheral quantitative computed tomography (pQCT), and high resolution pQCT, and found deficits in tibial cortical bone size and trabecular bone microarchitecture in 10 patients with ALGS compared to healthy controls [34]. Overall, the cause of bone fragility and increased fracture risk in ALGS is multifactorial, arising from chronic cholestasis, vitamin D deficiency, and intrinsic bone defects due to disrupted Notch signaling [34,35].

### 2.6. Renal Features

Renal involvement has been evident since the first reports in 1987 by Alagille et al. [2]. Based on large descriptive studies since then, prevalence of renal disease in ALGS has been estimated as 20–73% [2,9,10,11,12]. Due to this high frequency, experts have advocated it as a disease-defining feature, resulting in expansion of ALGS clinical criteria [13]. Both structural and functional abnormalities have been reported. The largest report to date included 466 patients with *JAG1* mutation positive ALGS showing 39% renal involvement; most commonly renal dysplasia (59%), followed by renal tubular acidosis (9.5%), and vesicoureteric reflux and urinary obstruction (8.2%). Progression to chronic kidney disease and renal transplantation in this setting is rare. Renovascular lesions with or without hypertension were reported elsewhere in 2–8% of patients [36].

It is important to note that pre-existing renal insufficiency may not resolve following LT in patients with ALGS, and in fact may worsen, as reported by a study utilizing a large North American liver transplantation database [37]. Ninety-one patients with ALGS were age-matched with 236 patients with biliary atresia (BA). Pretransplant glomerular filtration rate (GFR) was <90 mL/min/1.73 m^2^ in 18% of ALGS and 5% of BA patients, and had worsened to 22% and only 8% 1 year after LT, respectively. This highlights the developmentally abnormal kidneys in ALGS which are more vulnerable to nephrotoxic calcineurin immunesuppression and supports the use of renal-sparing immunosuppression protocols post-LT.

### 2.7. Vascular Features

Vascular involvement in patients with ALGS has long been unrecognized and can lead to life-threatening complications. Intracranial bleeding has been reported in approximately 15% of patients, and is responsible for death in 25–50% [9,10]. Clinical presentation is quite variable; ranging from silent cerebral infarcts found on screening brain imaging, to spontaneous fatal bleeding with or without symptoms [38,39]. Underlying central nervous system (CNS) vascular malformations, which are common in ALGS, clearly increase the risk of ischemic or hemorrhagic infracts. Emerick et al. screened 26 ALGS patients with brain magnetic resonance imaging (MRI) and angiography (MRA) [39]. Thirty-eight percent of patients had cerebrovascular abnormalities, of which half were asymptomatic. However, virtually 100% of patients with symptoms (which included hemiparesis, slurred speech, and generalized headache) had detectable lesions. Intracranial lesions reported in this study and others include narrowing/absence of internal carotid artery, aneurysms of the middle cerebral and basilar artery, moyamoya disease, subdural, subarachnoid, and epidural hemorrhage [10,38,39]. Due to the high prevalence and the deleterious effects of CNS vasculopathies, experts recommend routine screening with MRI/MRA at 8 years of age (the approximate age at which general anesthesia is not required for an MRI), and prior to any major surgery [40]. The frequency at which to repeat imaging, remains unclear since published data are scarce in this area.

Vascular involvement in ALGS can extend beyond the CNS and pulmonary vessels. Studies have identified subclavian, hepatic, celiac trunk, renovascular, and superior mesenteric anomalies [41,42,43]. In addition, aortic abnormalities such as coarctation and aneurysm have been reported and may be associated with intracranial lesions [38]. Due to the high prevalence of abdominal vascular anomalies that may complicate LT surgery, it is imperative to perform abdominal vascular imaging prior to LT to guide arterial reconstruction techniques. Kohaut et al. reported on 25 ALGS patients with almost 65% requiring arterial conduit reconstruction during transplant surgery [44].

## 3. Genetics of Alagille Syndrome

### 3.1. Gene Identification & Mutational Analysis

ALGS is an autosomal dominant disease with variable expressivity, caused by heterozygous mutations in either *JAG1* or *NOTCH2*. The vast majority of cases are due to *JAG1* mutations accounting for 94%, and *NOTCH2* mutations in additional 2–4% [5,45,46,47]. Sixty percent of patients harbour de novo mutations (i.e., sporadic). The remaining 40% inherit their mutation from a typically mildly affected parent [48,49]. 

After the first report in 1986 identified a deletion in the short arm of chromosome 20 in a child with a classical ALGS phenotype, investigators in 1997 discovered that *JAG1* mutations are causative for ALGS in four families [5,45]. Since then, 696 *JAG1* pathogenic variants have been described in patients with ALGS [46,50,51]. These are located in 26 exons that encode for the whole extracellular domain of *JAG1*, which are critical for *NOTCH2* binding and initiating the Notch signaling pathway (see below). The most common mutations are protein-truncating (75%) which include, in decreasing frequency, frameshift, nonsense, splice site, and gross deletion mutations. Non-protein truncating mutations include missense, in-frame deletion, duplication, translocation, and inversion [46,52]. 

*NOTCH2* was identified as a disease-causing gene after it was revealed that *JAG1/NOTCH2* double heterozygote mice developed an ALGS phenotype [53]. After screening a cohort of *JAG1* mutation negative patients in 2006, *NOTCH2* mutations were found in two probands [47]. Since then, 10 patients have been described with *NOTCH2* mutations [47,54], with the majority being missense mutations 77%. 

Overall, the observation that similar ALGS clinical phenotypes can be caused by different pathogenic mutations (protein-truncating, intragenic, whole gene deletions) suggest that haploinsufficiency of *JAG1* and *NOTCH2* is the primary mechanism for disease pathobiology [5,7,45,54], rather than a dominant negative mechanism.

### 3.2. The Notch Signaling Pathway & Bile Duct Development

The Notch pathway is a highly conserved fundamental signaling pathway responsible for cell-cell communication [55]. It is comprised of five ligands (*JAG1* and *2*, and Delta-like 1, 3, and 4), and 4 Notch receptors (1–4). Both *JAG1* and *NOTCH2* are single-pass transmembrane proteins with extracellular domains [46]. High-affinity binding is made possible through Delta-Serrate-Lag-2 (DSL), a critical extracellular domain of *JAG1*, and extracellular epidermal growth factor (EGF)-like repeats hosted by both *JAG1* and *NOTCH2* [56,57]. Ligand-receptor binding activates a *NOTCH2* intracellular domain, which translocates to the cell nucleus, thereby activating transcription of downstream genes, such as HES and HEY [58]. Thus, the Notch pathway is involved in cell fate determination and plays a crucial role in normal development [14]. 

Human embryological studies reveal that *JAG1* is highly expressed in organs that are typically affected in patients with ALGS, such as the heart, kidneys, vessels, skeleton, and eyes [59]. This highlights the importance of the Notch pathway in the development of the organs involved in ALGS. In particular, the presence of bile duct paucity in *JAG1*-mutation positive ALGS patients, has revealed the crucial role of Notch signaling in the development of intrahepatic bile ducts. It is beyond the scope of this article to review this here; however, it is clear from mice data that specifically *JAG1-NOTCH2* interactions are crucial for intrahepatic bile duct development [60,61].

## 4. Diagnostic Testing

The biochemical profile of patients with ALGS reflects biliary damage and cholestasis. Markers of cholestasis (serum bile acids, bilirubin, cholesterol, *γ*-glutamyltransferase (GGT), and alkaline phosphatase) are often strikingly elevated and almost always exceed that of hepatocellular injury (alanine and aspartate aminotransferase) [15]. GGT may, however, be normal, and therefore should not defer further testing if there is high index of suspicion for ALGS [12]. Cholestasis often spontaneously improves in patients during childhood, and is accompanied by a reduction in pruritus and xanthomas [18].

Due to biliary tree hypoplasia, liver ultrasonography may show small or absent gallbladder in 28% of patients [12]. Hepatic regenerative nodules have been reported in 30% of patients and can be confused with hepatocellular carcinoma. They are distinguishable biochemically by normal alpha-fetoprotein, and radiologically by their central location, isoechoic texture to surrounding liver, and absence of invasion of portal venous structures on MRI [62,63,64].

Lastly, in any patient where ALGS is suspected, formal echocardiography, dedicated vertebral radiography, slit-lamp examination of the eyes, and renal ultrasonography with doppler should be performed. Brain MRI/MRA is recommended in patients with ALGS with neurologic symptoms or for screening later in childhood.

### 4.1. Liver Histopathology

With advancements in molecular diagnostics, a liver biopsy is no longer required to diagnose ALGS, but remains an integral part of clinical diagnosis if molecular testing is not available in a timely fashion, and to differentiate between ALGS and BA [14]. Bile duct paucity remains the hallmark of ALGS and was once an absolute requirement for diagnosis. It is assessed by calculating the interlobular ducts to portal tracts ratio, with normal being 0.9–1.8 [3], and is diagnosed if <0.9 in full term infants. As the number of ducts to portal tracts decrease overtime, the ratio is typically < 0.5–0.75 in older infants [3,65]. Bile ductules that are usually peripherally located are not included in this ratio. It is important to have an adequate number of evaluated portal tracts to arrive at a precise ratio. When wedge biopsies were historically used, Alagille et al. suggested that 20 portal tracts should be evaluated [3]. However, 6–10 portal tracts are usually sufficient at present with needle biopsies [65,66,67].

To date, bile duct paucity has been reported in approximately 89% of patients with ALGS [2,8,9,11]. It is more commonly found in children > 6 months of age as reported by Emerick et al. where paucity was found in only 60% of children < 6 months compared to 95% in > 6 months [10]. Factors leading to the paucity progression (which mirror severity of clinical hepatic phenotype) are unknown, but hypotheses include postnatal ductal destruction, lack of development of terminal branches of the bile ducts, and/or differential maturation of portal tracts [10,14]. 

Other reported histological features include occasional ductular proliferation that is typically associated with portal inflammation, and giant cell hepatitis due to cholestasis (which may mimic BA). Regenerative liver nodules if found, show preserved ductal architecture, lesser degrees of fibrosis and relative preservation of interlobular bile ducts compared to the background cirrhotic liver [62,63,64].

### 4.2. Genetic Testing

Genetic testing in clinically defined patients with ALGS reveals a disease-causing mutation in almost 95%. Since most ALGS-associated mutations are found in *JAG1*, Sanger sequencing of all 26 exons and adjacent intronic regions will identify 85% of *JAG1* pathogenic variants. If no mutation is found, large deletion duplication analysis using multiplex ligation-dependent probe amplification (MLPA), chromosomal microarray (CMA), or fluorescence in situ hybridization (FISH) will successfully identify an additional 9% of mutations [46,68]. If no *JAG1* mutation is found, sequencing of the 34 exons encoding for the *NOTCH2* gene should be carried out, which identifies 2–3% of additional mutations. MLPA, FISH, and CMA are typically not carried out since no large deletions of *NOTCH2* have been reported. [46,68]. For practical reasons and cost-effectiveness, simultaneous testing of both genes is now carried out by commercially available next generation sequencing (NGS) panels.

In the remaining 4% of cases who meet clinical criteria for ALGS but do not have an identified mutation, novel gene discoveries may be found via whole exome, or genome sequencing. Alternative diagnoses should be sought in mutation-negative ALGS patients, especially if presenting with the less penetrant features of the disease [69,70].

## 5. Diagnostic Challenges

### 5.1. Genotype-Phenotypic Variability

Data on the clinical phenotype of patients with *JAG1*-associated ALGS are vast, compared to *NOTCH2*, as only a small number of patients have been reported with *NOTCH2* mutations. Skeletal anomalies and characteristic facies seem markedly less penetrant features in *NOTCH2* probands than *JAG1*; 10% vs. 64%, and 20% vs. 97%, respectively [7,54]. Additionally, there is trend towards less cardiac involvement in *NOTCH2* probands compared to *JAG1* (60% vs. 100%) [54]. A few patients with large deletions of chromosome 20p12.2 (the ALGS critical region) greater than 5.4Mb have been described who have clinical features in addition to classic ALGS such as developmental delay and hearing loss [71].

The extreme variable expressivity of ALGS and the presence of more than 700 different pathogenic variants in *JAG1* and *NOTCH2* probands (including whole gene deletion and protein-truncating mutations) conceptually favors a genotype-phenotype correlation. However, no correlation has been found in large cohort studies [46]. The same mutation can be associated with wide range of clinical findings within families, including monozygotic twins [49,54,72]. This suggests the presence of genetic modifiers.

Genetic modifiers that could attribute to variable expressivity include modifications to glycotransferase of *JAG1* and *NOTCH2* protein, a process that normally maintains receptor-ligand binding and proper protein folding [73,74,75]. Mice heterozygous for Fringe genes that glycotransferase *JAG1* had increased bile duct proliferation and remodeling, suggesting that Fringe genes can modify the liver phenotype [75]. A similar finding of decreased bile duct paucity was found in mice heterozygous for *JAG1* and loss of *Rumi* glycotransferase [76]. This suggested that humans with a known *JAG1* mutation and overexpression of POGLUT1 (the human homolog for *Rumi*) may have worse liver disease. In a genome wide approach to the identification of ALGS liver disease modifiers, in comparison to patients with mild liver disease, those with severe liver disease were found to have single nucleotide polymorphism located upstream of Thrombospondin2 (THBS2), implicating this gene as a potential modifier. THBS2 encodes an extracellular matrix protein expressed in murine bile ducts and inhibits *JAG1-NOTCH2* binding [77]. The identification of potential genetic modifiers of the ALGS hepatic phenotype remains an active area of study.

With only limited data emerging regarding potential genetic modifiers of the Alagille phenotype, it is enticing to consider if epigenetic modification of Notch signaling could be a relevant factor. Unfortunately, this phenomenon has been rarely studied in the context of ALGS. One study suggested that prohibitin-1 may modulate cholestatic liver injury in ALGS by regulating histone deacetylase 4 (HDAC4) [78]. However, these data have not been substantiated further but remain intriguing.

In summary, it is not possible to predict ALGS disease burden due to the absence of genotype-phenotype correlations and the extreme variable phenotypic variability in patients with ALGS. This is of utmost importance to highlight during genetic counseling and in the event of prenatal genetic testing.

### 5.2. Bile Duct Paucity

A source of diagnostic dilemma to clinicians is the presence of bile duct paucity in patients who otherwise do not fit the clinical ALGS description or are non-syndromic. Bile duct paucity is not pathognomonic for ALGS and can be found in genetic disorders (Trisomy 21), metabolic disorders (α_1_-antitrypsin deficiency and cystic fibrosis), infections (congenital cytomegalovirus, rubella, and syphilis), immune disorders (hemophagocytic lymphohistiocytosis), and secondary to drug-induced vanishing bile duct syndrome (Table 1) [15,79]. 

The presence of bile duct paucity in ALGS patients varies with age, and is absent in 40% of children < 6 months [10]. In addition, due to normal continued postnatal ductal development and remodeling, preterm (less than 38 weeks) and small for gestational age infants have physiological immaturity of bile ducts, and therefore appropriately have a bile duct to portal space ratio < 0.9 (i.e., physiological bile duct paucity) [80,81]. These situations create diagnostic uncertainty to hepatologists and pathologists, particularly if BA is in question, where bile duct paucity can also be seen in almost 10% of patients [67]. Thus, the histopathologic features of ALGS and BA can overlap. Mutational analysis is now commercially available within 3–4 weeks; however, this timeframe may still be too long in an infant in whom a time-sensitive Kasai procedure for BA may be required. Occasionally, a repeat liver biopsy is warranted though more frequently, cholangiography is warranted.

### 5.3. Cholangiography

ALGS can often be misdiagnosed as BA, due to significant overlap of biochemical, histologic, and imaging features. Due to the time-sensitive nature of Kasai procedure, and improved outcome in BA if performed before 60 days of age [82], patients with ALGS may undergo Kasai procedure, which is associated with poor outcome [83]. Hepatobiliary scintigraphy (HIDA) and operative cholangiography are utilized to aid with this distinction but even these findings may overlap between ALGS and BA.

Excretion of nuclear tracers in HIDA scans into the duodenum effectively excludes BA. However, non-excretion is quite common in ALGS despite adequate hepatic radiotracer uptake. Emerick et al. reported on 36 patients with ALGS where more than 60% had no excretion of isotope into the bowel after 24 h [10]. 

Operative cholangiography is the gold standard to diagnose BA, and to evaluate the intra- and extrahepatic biliary tree, but may be misleading if interpreted without taking into account other clinical and diagnostic data. In ALGS, cholangiograms may commonly show non-communication proximally due to small intra- and extrahepatic ducts [10]. Thirty-seven percent have no visualization of the proximal extrahepatic ducts (hepatic duct to hilum), and an additional 37% are hypoplastic. These findings that can mimic BA, can lead to a Kasai procedure being performed, which is likely worsens hepatic outcomes in ALGS. Kaye et al. compared 19 ALGS patients who underwent a Kasai procedure to 36 matched ALGS controls [83]. Despite sharing similar biochemical findings at presentation between groups, the Kasai cohort had higher rates of LT (47% vs. 14%) and mortality (32% vs. 3%), suggesting that a Kasai is not a marker of liver disease severity, but that the procedure itself is harmful and responsible for worse outcome. Similar negative outcomes were observed in a recent systematic review and meta-analysis [84].

In an effort to avoid a Kasai procedure in ALGS when nuclear scans, cholangiographic, and histologic studies are inconclusive, time permitting, clinicians should opt for expedited mutational analysis for *JAG1* and *NOTCH2* simultaneously which may be available in 3–4 weeks. Additionally, recent data support the utility of serum matrix metalloproteinase-7 as a biomarker for diagnosing BA, performing superiorly against other causes of neonatal cholestasis, including ALGS [85].

## 6. Management of Alagille Syndrome

Treatment for patients with ALGS is supportive and aimed at optimizing nutrition and managing complications related to cholestasis, such as FSVD and pruritus.

### 6.1. Nutrition and FSVD

The etiology of growth failure in ALGS is multifactorial and includes inadequate intake, fat malabsorption due to cholestasis, and cardiac disease [86]. Although children with ALGS, have normal resting energy expenditure [87], they require 25% additional recommended daily allowance due to cholestasis [88], and may even require more for catch up growth if malnutrition is severe [15]. Patients should be encouraged to consume calorie dense food, especially medium chain triglycerides-rich foods or formula, since they do not require micellar formation for absorption. Nasogastric or gastrostomy tube feeding should be considered in children unable to meet their caloric needs and is often necessary in cholestatic children.

Cholestasis-specific formulations exist for fat-soluble vitamin (FSV) supplementation (e.g., DEKAs), which may help with medication compliance and cost, especially in patients with multiple FSVD. If only generic multivitamin preparations are available, then individual supplementation of FSV is preferred.

### 6.2. Medical Management of Pruritus

Therapies aiming at decreasing total body bile acid load are typically effective for pruritus. However, there are likely other mechanisms underlying pruritus, as serum bile acid levels do not always correlate with itching severity [15]. Pruritus treatment in ALGS follows a step-by-step approach, as outlined in Table 2. Antihistamines are used for mild cases. They are rarely used as single agents due to their short-lived effect [89]. Ursodeoxycholic acid (UDCA) promotes bile excretion and makes it more hydrophilic and is used in most cholestatic children with ALGS. Other available therapies include rifampin, bile salt-binding agents (cholestyramine), opioid antagonists, and selective serotonin re-uptake inhibitors (SSRI) such as Sertraline [15,90]. Cholestyramine disrupts the enterohepatic circulation and reduces total body bile acids by preventing re-uptake in the terminal ileum. Due to its poor taste, interference with absorption of food, medications, and FSV, it is of limited use in clinical practice [89]. Rifampin is thought to 6-hydoxylase bile acids making them less pruritogenic [91] and excretable by the kidneys [92]. Almost 50% of patients treated with Rifampin report good improvement in pruritus [89]. Naltrexone, an opioid antagonist that blocks mu receptors, which are upregulated in cholestasis [93], has been associated with at least minimal improvement in most children with ALGS [89]. However, symptoms of opioid withdrawal syndrome, such as diarrhea and irritability, occur in almost 30% of patients. Although sertraline, a selective serotonin reuptake inhibitor (SSRI) has been effective in treating adults with cholestatic pruritus [94], its mechanism of action is unknown. Limited pediatric data are available supporting its use as additional therapy for pruritus [95].

### 6.3. Surgical Management of Pruritus

In ALGS patients with pruritus refractory to medical therapy, surgical procedures targeted at interrupting the enterohepatic circulation should be considered. Since bile duct hypoplasia associated with ALGS can result in less bile reaching the bowel, these procedures are generally less effective than in other causes of cholestasis (such as progressive familial intrahepatic cholestasis). Partial external biliary diversion (PEBD), where the gallbladder is drained externally via a jejunal conduit, is the most commonly performed procedure [96]. Wang et al. reported improvement in total serum cholesterol, pruritus severity, and xanthomas in 20 ALGS patients who have undergone PEBD [97]. Other less commonly performed procedures include ileal exclusion and internal biliary diversion.

### 6.4. Liver Transplantation

The indications of LT in ALGS are typically multifactorial but can be broadly classified as end-stage liver disease due to progressive cholestasis (malnutrition refractory to nutritional therapy, intractable pruritus, and bone fractures) and/or end-stage liver disease with portal hypertension and complications, such as ascites and variceal bleeding [98]. When assessing candidacy, careful consideration should be sought for the multisystemic involvement; cardiac, renal, and vascular disease. As mentioned previously, patients should undergo MRI/MRA of the brain and computed tomography (CT) imaging of the abdomen, and echocardiogram when being assessed for transplant. Renal-sparing immunosuppression protocols should be used. 

When considering living related transplantation, it is important to emphasize that donors with *JAG1* and/or *NOTCH2* mutations should be avoided as they may have unrecognized liver disease. Therefore, all potential related donors should have a comprehensive clinical assessment, genetic screening for the known mutation in the proband, abdominal imaging for vascular anomalies, and potentially liver biopsy [14,98].

Among ALGS children presenting with cholestasis, LT is required in almost 75% by the age of 18 [18]. ALGS comprises 4% of all pediatric LT cases combined [99]. The largest multicenter study of post-transplant ALGS data utilizing the United States United Network for Organ Sharing database described outcomes in 461 children [99]. One- and 5-year survival were 82% and 78%, respectively. Death in the first month was higher in ALGS than BA, and overall death from graft failure, neurologic, and cardiac complications were higher in ALGS. Another multicenter study evaluating transplant outcome data on 91 children with ALGS over a 14-year period also showed similar survival outcome measures (One- and 5-year survival 83% and 78% respectively) [37], and noted clustering of death in the first 30 days once again. This may be explained by the multisystemic involvement of ALGS and burden of associated comorbidities.

## 7. Advancement in Management in Alagille Syndrome

### 7.1. IBAT Inhibitors

The concept of molecular therapy with IBAT inhibitors is similar to that of biliary diversion procedures; reduction of the total bile acid pool size via inhibition of enterohepatic circulation, results in mitigating the toxic effects of bile acids on the liver and improvement of cholestasis [100,101]. Located on the apical membrane of ileal enterocytes, IBATs actively transport conjugated bile acids from enterocytes, which are then exported into the portal system via different mechanisms, facilitating return to the liver [102]. As a result, more than 90% of intestinal bile acids are reabsorbed in healthy individuals [102,103]. Currently two drugs in this class are under study for pruritus in children with cholestasis—Maralixibat and Odevixibat—though at this time there are more available data for the former in the study of ALGS.

Efficacy of Maralixibat, an IBAT inhibitor, has been evaluated in phase 2 trials in patients with ALGS. The ITCH trial evaluated 37 patients with ALGS in a placebo-controlled randomized trial [104]. Although the pre-specified primary endpoints were not met in this study, a reduction in pruritus, as measured by caregiver observation a validated scale (ItchRO), was more common in the Maralixibat treated group as compared to the placebo group. Maralixibat was safe with comparable adverse events between groups. The ICONIC trial evaluated 31 patients with ALGS in a multicentered trial using a randomized drug-withdrawal study design (though these data have only been presented in abstract form, to date) [105]. Serum bile acids levels fell, as expected, on Maralixibat treatment; however, during the randomized drug withdrawal period, bile acid levels in the placebo group returned to baseline, and subjects had significantly higher ItchRO scores. [105].

These preliminary studies show that IBAT inhibitors hold promise as future treatments for pruritus that may potentially also prove to be hepatoprotective. Continued investigations are warranted to explore their therapeutic effect on the natural history of cholestatic disease in ALGS.

### 7.2. Cholangiocyte Regeneration

The cholangiopathy of ALGS involves defects in cholangiocyte specification, differentiation and morhpogenesis, making this pathobiologic process subject for investigational cell rescue and/or tissue regeneration. Similar to stem cell-mediated organ regeneration, cellular transdifferentiation is a process of complete and stable change in cell identity. This makes it an attractive system to utilize in repairing the defective biliary system in ALGS [106], by potentially harnessing the ability of hepatocytes to transdifferentiate into cholangiocytes.

In a recent important study, transdifferentiation was explored in an ALGS mouse model made by *NOTCH* deletion, showing severe cholestasis and lacking peripheral bile ducts [106]. At postnatal day 120, newly formed peripheral bile ducts were detected, with cholangiocytes harboring markers indicative of hepatocyte origin. Hepatocyte-derived peripheral bile ducts (HpBDs) were found to be contiguous with the extrahepatic biliary system and were effective in draining bile, evident by normalization of total bilirubin [106]. HpBDs showed signs of cholangiocyte maturity and authenticity and expressed markers of biliary differentiation, indicating that they were not merely hepatocyte-derived metaplastic biliary cells. This transdifferentiation was not only limited to immature hepatocytes, but also seen in murine adult and transplanted hepatocytes [106]. This signifies that hepatocytes can form peripheral bile ducts de novo and can provide normal and stable biliary function. Further investigations in this report led to the discovery that TGFβ is responsible for hepatocyte transdifferentiation and morphogenesis in HpBD formation [106]. This was also identified in regenerative nodules in adult ALGS patients that stained positive for cytokeratin-7 and contained peripheral bile ducts, suggestive that this mechanism is active in humans with ALGS. This study not only highlights the significance of hepatic plasticity and cellular transdifferentiation, but also emphasizes the utility of therapeutic hepatocyte transplantation, and targeting TGFβ induction as future treatment strategies in ALGS-related cholestasis.

### 7.3. Stem Cell Applications in ALGS

The rationale for using stem cell technology to model and perhaps treat biliary diseases is powerful and includes reasons, such as limited access to human biliary tissue, lack of physiological responses in cultured cholangiocytes, and the inability of murine models to fully recapitulate human biliary disease [107]. Induced pluripotent stem cells (iPSCs) are generated through reprogramming mature human somatic cells to a pluripotent state [108]. iPSCs have the potential to differentiate into any germ layer in vitro, which when utilizing unique protocols can be directly differentiated into almost any cell type, including cholangiocytes. Cholangiocytes that express mature biliary markers and demonstrate biliary functions have been successfully differentiated by a number of groups [109]. Furthermore, iPSC-derived cholangiocytes have been shown to recapitulate disease features of cystic fibrosis-related cholangiopathy [110].

IPSC technology has yielded robust results in modeling ALGS liver pathology when comparing iPSCs-derived hepatic organoids from two ALGS patients and three controls [111]. ALGS patients had marked reduction in cholangiocyte markers (such as CK-7 and GGT) and 90% of structures formed were vesicles rather than intact organoids, as seen in controls. Furthermore, when genome editing permitted mutation reversal in ALGS iPSCs, organoids formed well organized bile-duct forming structures [111]. This highlights how iPSCs can revolutionize our understanding of disease pathophysiology and how they can be utilized for future drug discovery in ALGS. The clinical applications of iPSCs, however, such as cellular transplantation, remain a concern at present due to their genomic instability and malignant potential [112,113]. Further research is required to establish the safety of iPSCs for patient cellular therapy.

## 8. Conclusions

ALGS is a complex disease with significant inter and intrafamilial variable expression that poses significant diagnostic challenges and requires high index of suspicion for diagnosis. Although the road for future targeted therapies is promising, the lack of genotype-phenotype correlation and absence of clinical and molecular predictors of disease outcome is a cause of significant uncertainty to clinicians and families. The recent establishment of the Global ALagille Alliance (GALA) Study, may help overcome these limitations [114]. The collective effort of this international collaborative consortium from more than 20 countries, will deepen our understanding of ALGS, its natural history and disease burden.

## Figures and Tables

**Figure 1 diagnostics-10-00907-f001:**
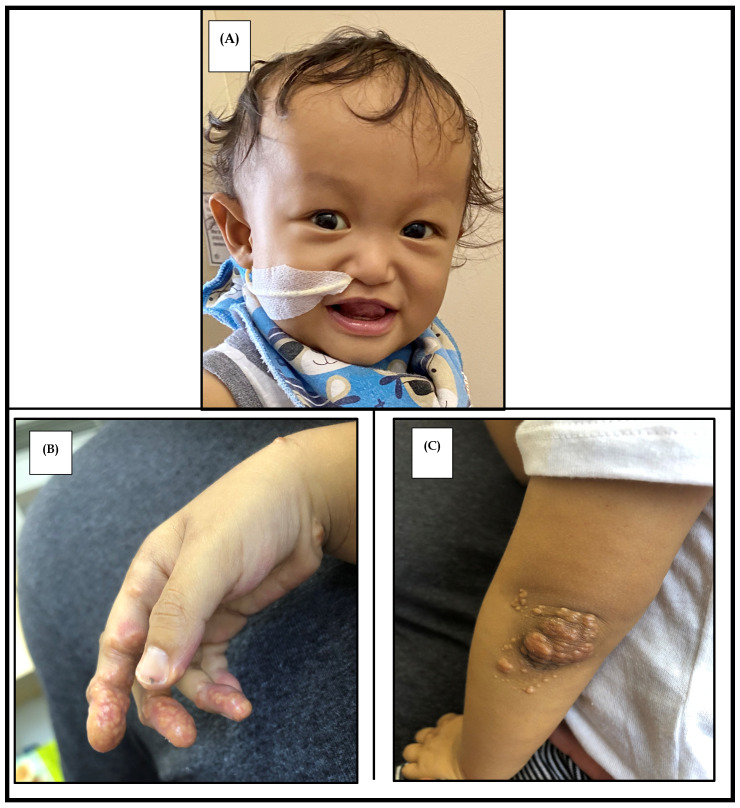
Clinical features of alagille syndrome. (**A**) Characteristic facies with prominent forehead, hypertelorism, straight nose with a bulbous tip, and a pointed chin. Parental consent was obtained for use of this photograph. (**B**) Cutaneous xanthoma of the palmar surface of the hand. (**C**) Xanthoma on the elbow.

**Table 1 diagnostics-10-00907-t001:** Differential Diagnosis of Bile Duct Paucity.

Disease Type	Cause
Genetic	Alagille syndrome
	Trisomy 21
	Williams syndrome
	Peroxisomal disorders
Metabolic	α_1_-antitrypsin deficiency
	Cystic fibrosis
	Panhypopituitarism
Infections	Congenital cytomegalovirus
	Congenital rubella
	Congenital syphilis
Immune/Inflammatory disorders	Hemophagocytic lymphohistiocytosis
	Sclerosing cholangitis
	Graft-versus-host disease
	Chronic allograft rejection
Other	Drug-associated vanishing bile duct syndrome
	Biliary atresia (late finding)

**Table 2 diagnostics-10-00907-t002:** Pharmacological step-up therapy of cholestatic pruritus in Alagille syndrome.

Medication Class	Medication	Side Effect Profile
**1st line:** Choleretics	Ursodeoxycholic acid	Generally safe; Diarrhea, abdominal pain, vomiting
Bile salt-binding agents	Cholestyramine	Constipation, abdominal pain, worsening FSVD, poor palatability
**2nd line:** Bile acid hydroxylation	Rifampin	Red discoloration of bodily fluids (sweat, tears), vomiting, hepatitis, idiosyncratic hypersensitivity reaction
**3rd line:** Opioid antagonists	Naltrexone	Limited data; abdominal pain, nausea, irritability, diarrhea
**4th line (not yet approved by regulators):** Intestinal bile acid transport (IBAT) inhibitors	Maralixibat Odevixibat	Limited data; vomiting, diarrhea, abdominal pain, rash, hepatitis, FSV deficiencies
**Adjunctive therapy:**		
Antihistamines	Diphenhydramine	Drowsiness
SSRI	Sertraline	Limited data; agitation, alopecia and drug eruption, vomiting, hypertension

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
