# Peer review of "Alagille Syndrome: Diagnostic Challenges and Advances in Management"

_diagnostics, 2020, doi:10.3390/diagnostics10110907_

Round 1
Reviewer 1 Report
Good review!!!
Author Response
Thank you for your comment. We hope this review will add to the body of literature on Alagille syndrome.

Reviewer 2 Report
This is very well written and thorough review on Alagille Syndrome with a clinical focus. The authors have covered well the features of the disease, genetics and treatments.
With the lack of genotype-phenotype relationships in certain cases, can the authors touch base on known epigenetic modulators/regulators that could be therapeutically targeted in this disease.
In the management section, the authors should add a section on how iPSC technology can be utilized for this disease.
Author Response
Point 1: With the lack of genotype-phenotype relationships in certain cases, can the authors touch base on known epigenetic modulators/regulators that could be therapeutically targeted in this disease.
Response 1: Many thanks for this interesting comment. Unfortunately, there are very few data that address epigenetic modulation in Alagille syndrome, however, as suggested we have addressed this topic in a short paragraph on page 10 and included a relevant reference [Barbier‐Torres, L.; Beraza, N.; Fernández‐Tussy, P.; Lopitz‐Otsoa, F.; Fernández‐Ramos, D.; Zubiete‐Franco, I.; Varela‐Rey, M.; Delgado, T.C.; Gutiérrez, V.; Anguita, J. Histone deacetylase 4 promotes cholestatic liver injury in the absence of prohibitin‐1. Hepatology 2015, 62, 1237-1248.]
Point 2: In the management section, the authors should add a section on how iPSC technology can be utilized for this disease.
Response 2: We have added a separate section on page 16 to address this topic. We explain the rationale behind utilizing this technology, as well as properties unique to iPSCs, particularly in its ability to model ALGS. We lastly touch on the current limitations of using this technology in clinical care. Six references were added accordingly.
